# Development of a Simultaneous Process of Surface Modification and Pd Nanoparticle Immobilization of a Polymer Substrate Using Radiation

**DOI:** 10.3390/nano12091463

**Published:** 2022-04-25

**Authors:** Naoto Uegaki, Satoshi Seino, Yuji Takagi, Yuji Ohkubo, Takashi Nakagawa

**Affiliations:** Graduate School of Engineering, Osaka University, 2-1, Yamada-oka, Suita 565-0871, Osaka, Japan; yuji5553@gmail.com (Y.T.); okubo@upst.eng.osaka-u.ac.jp (Y.O.); nakagawa@mit.eng.osaka-u.ac.jp (T.N.)

**Keywords:** radiation, Pd nanoparticles, adhesion strength

## Abstract

Pd nanoparticles were immobilized on an acrylonitrile–butadiene–styrene copolymer (ABS) substrate using ionizing radiation. The samples were prepared by irradiating plastic zipper packs containing ABS substrates and a Pd(NO_3_)_2_ aqueous solution with a high-energy electron beam (4.8 MeV). Pd nanoparticles immobilized on the ABS substrate surfaces were observed using scanning electron microscopy (SEM). The chemical state of Pd was found to be coordinated to a carbonyl group or a metallic state by X-ray photoelectron spectroscopy (XPS) measurements. The peel strength of the Cu film on the Pd/ABS samples was 0.7 N/mm or higher. This result shows that the prepared Pd/ABS samples have high adhesion strength, despite not undergoing treatments such as etching with chromic acid. This method is expected to immobilize metal nanoparticles, not only on plastic plates but also on various other materials.

## 1. Introduction

Plating is a surface treatment technology that can add functions, such as product decoration, wear resistance, rust prevention, corrosion prevention, and electrical conductivity. Plating can be subdivided into electroless plating and electroplating. Electrolytic plating is a method of forming a metal film by passing an electric current through a product immersed in the desired metal salt as a cathode. Electroless plating, on the other hand, is a method of forming the desired metal film on a product using a reducing agent. Therefore, non-conductive materials, such as plastic substrates and ceramics, require electroless plating, followed by electroplating.

The process for electroless plating typically includes three steps: (i) surface preparation; (ii) surface activation or surface seeding with the metal, which is the catalyst for the next step; and (iii) electroless plating bath, which consists of a redox reaction between metallic ions and a strong reducer contained in the same solution. Pd is usually employed as a catalyst to initiate electroless plating and activate metal reduction onto the polymer [1,2,3].

The acrylonitrile–butadiene–styrene copolymer (ABS) is currently the most widely used resin substrate for plating because of its mass productivity, dimensional stability, impact resistance, chemical resistance, and high processability. In conventional processes, a chromic acid–sulfuric acid solution is used for the surface treatment of ABS substrates. Chromic acid dissolves the polybutadiene phase in ABS, which increases the surface roughness and significantly improves mechanical adhesion [4,5,6,7]. However, the hexavalent chromium ion contained in chromic acid has a high environmental impact and is strictly regulated by the RoHS Directive and REACH. Therefore, various electroless plating methods that do not require hexavalent chromium have been proposed as alternatives to the conventional surface treatment that uses a chromic acid–sulfate solution [8,9,10,11,12,13,14,15].

Therefore, our research group has proposed a new electroless plating method based on the nanoparticle synthesis method using radiation [16]. In this method, metal nanoparticles are synthesized via a chemical reaction involving irradiation. This method can be used to immobilize metal nanoparticles on a variety of support surfaces and has been developed to synthesize catalytic materials such as PtRu/C [17]. Ohkubo et al. reported that Pt nanoparticles were immobilized on ABS substrates using this method [18]. In addition, immobilized Pt nanoparticles have been reported to act as catalysts for H_2_O_2_ decomposition.

In this study, the direct immobilization of Pd nanoparticles on ABS substrates using radiochemical reactions was demonstrated. If Pd nanoparticles can be directly immobilized on ABS substrates, the impact on the environment will be minimal.

## 2. Materials and Methods

Pd nanoparticles were immobilized on a polymer substrate using radiation. The elemental processes of metal ion reduction have been described in previous studies [19]. A description of the method for the radiolytic synthesis of Pd/ABS follows.

### 2.1. Immobilization of Pd on Substrates

The chemicals used were palladium (II) nitrate (Pd(NO_3_)_2_; Tanaka Kikinzoku Kogyo K.K., Tokyo, Japan) and 2-propanol (CH_3_CH(OH)CH_3_; Wako, Osaka, Japan). Commercial ABS plates ((CH_2_CH(C_6_H_6_)·CH_2_CHCHCH_2_·CH_2_CH(CN))*_n_*, AS-ONE, Osaka, Japan) were used as substrates. Before use, the substrates were washed with 2-propanol and pure water, then dried at room temperature. Plastic zipper packs (Uni-Pack E-4, Seisannipponsha Ltd., Tokyo, Japan) were used as reaction vessels.

An aqueous solution of Pd(NO_3_)_2_ (1 mM) in 2-propanol (0, 1, and 10 vol%) was used as the precursor solution. The substrates (50 × 50 × 3 mm^3^) were immersed in 20 mL of the precursor solution. Plastic zipper packs containing the substrates and precursor solution were lined up on the irradiation tray. The samples were irradiated with a high-energy electron beam (4.8 MeV) using a dynamitron accelerator (SHI-ATEX, Tokyo, Japan). The surface dose was maintained at 20 kGy. It is assumed that electron beam irradiation induces chemical reactions generating Pd nanoparticles and introducing functional groups on the surface of ABS substrate simultaneously. Pd nanoparticles formed by the irradiation were expected to be immobilized on the ABS substrate surface. After irradiation, the substrates were removed from the solution and washed with pure water using an ultrasonic cleaner for 5 min. The washed samples were dried at room temperature. The obtained samples were denoted Pd/ABS 0 vol%, Pd/ABS 1 vol%, and Pd/ABS 10 vol%, respectively. All obtained samples were characterized by scanning electron microscopy (SEM), inductively coupled plasma atomic emission spectroscopy (ICP-AES), and X-ray photoelectron spectroscopy (XPS).

### 2.2. Material Characterization

The amount of Pd immobilized on the substrates was analyzed by ICP-AES (ICPE-9000, Shimadzu, Kyoto, Japan). The surfaces of the substrates were washed with aqua regia (HCl:HNO_3_ = 3:1) to dissolve the Pd nanoparticles. The surfaces of the irradiated samples were characterized by SEM-EDX (JSM-7001F, JEOL Ltd., Tokyo, Japan) and XPS (Quantum 2000, ULVAC-PHI, Kanagawa, Japan). Prior to the SEM observation, the surfaces of the resin plates were coated with Os (OPC60A, Filgen, Aichi, Japan). The surface chemical state was analyzed via XPS using an Al Kα X-ray source operating at 15 kV. A C1*s* level of 284.6 eV was used as an internal standard to correct for the peak drift. All high-resolution XPS peaks were fitted using the XPSPEAK41 software.

### 2.3. Electroless Plating Process

The electroless plating process described in this study is divided into two main steps.

(i) Acceleration step: In this stage, the immobilized Pd particles were converted to Pd^0^, increasing their activity. Before acceleration, wettability was imparted by dipping the substrates in a 5.0 vol% Thru-Cup MTE-1-A (C. Uyemura & Co., Ltd., Osaka, Japan) aqueous solution at 50 °C for 2 min. After pretreatment, the samples were washed with water at 50 °C for 1 min and rinsed with water at room temperature. The accelerators used were ALCUP Reducer MAB-4-A, MAB-4-C and MRD-2-C (C. Uyemura & Co., Ltd., Osaka, Japan). The Pd-immobilized sample was accelerated using the following composition: 1.0 vol% MAB-4-A, 5.0 vol% MAB-4-C, and 1.0 vol% MRD-2-C in an aqueous solution. 

(ii) Electroless Cu-plating step: The surface-activated Pd/ABS substrates were placed in electroless Cu-plating baths. The conditions of the aqueous solution in the plating bath were as follows: 10 vol% Thru-Cup PEA-40-M, 6.0 vol% Thru-Cup PEA-40-B, 3.5 vol% Thru-Cup PEA-40-D, and 2.3 vol% 37% formaldehyde. The temperature of the plating bath was set to 34 °C. The electroless Cu-plated ABS substrates were then rinsed with distilled water and dried in warm air. Subsequently, they were subjected to thermal treatment at 80 °C for 60 min.

### 2.4. Electroplating Process

Electroplating is necessary for measuring the adhesion strength of electroless Cu plating on ABS in a 90° peel test. Acid degreasing and acid activity treatment were performed as pretreatments. For acid degreasing, the substrates were immersed in a 10 vol% DP-320 (Okuno Chemical Industries Co., Ltd., Osaka, Japan) aqueous solution at 40 °C for 5 min. For the acid activity treatment, the substrates were immersed in 1.82 mol/L H_2_SO_4_ at room temperature for 30 s. Electroplating was carried out in a 0.5 vol% Top Lucina 2000MU (Okuno Chemical Industries Co., Ltd., Osaka, Japan), 0.05vol% Top Lucina 2000A (Okuno Chemical Industries Co., Ltd., Osaka, Japan), 1.25 mol/L CuSO_4_, 0.5 mol/L H_2_SO_4_, and 0.0175 vol% conc. HCl, at a temperature of 25 °C. The cathode current density was maintained at 3 A/dm² for 1 h.

### 2.5. Adhesion Strength Test

The adhesion strength was determined by measuring the 90° peel adhesion strength using an AG-Xplus (Shimadzu, Kyoto, Japan) with a 1 kN load cell. All measurements were carried out at a width of 10 mm, peel-off speed of 50.0 mm/min, and a constant temperature of 24 °C. The adhesion strength was reported as the average of three sample measurements.

## 3. Results

### 3.1. Characterization of Pd/ABS

Figure 1a–f shows SEM images of the Pd/ABS samples. EDX analysis revealed that the nanoparticles were observed as white spots consisting of Pd. In addition, Pd nanoparticles with diameters of approximately 50–200 nm were observed in all samples. The amounts of Pd nanoparticles immobilized on the resin substrates were 2.4, 0.3, and 3.2 μg/cm^2^ with experimental conditions of 0, 1, and 10 vol% 2-propanol, respectively. As shown in ICP-AES analysis, loading amount of Pd on ABS did not depend on the concentration of 2-propanol. These results clearly indicate that the Pd nanoparticles were successfully immobilized in the presence and absence of 2-propanol. 

Figure 2a,b shows the XPS analysis results for the Pd/ABS samples. The XPS peaks of Pd3*d* synthesized with 10 vol% 2-propanol were 335.2 eV and 340.5 eV, which correspond to the metallic Pd [20]. In contrast, the peaks obtained with the 0 vol% 2-propanol were 339.2 and 344.5 eV (Figure 2a). These peaks are attributed to the state in which Pd is coordinated to the oxygen of the carbonyl groups [21]. For the samples synthesized with 1 vol% of 2-propanol, peaks corresponding to the metallic state and to the coordination of the carbonyl groups with oxygen were observed. These results indicate that 2-propanol, used as a radical scavenger, significantly affects the chemical state of the immobilized Pd nanoparticles.

After irradiation, the peak intensities of O1*s* increased for all the samples (Figure 2b). As polymers constituting the ABS substrates do not contain oxygen, it was assumed that the oxygen-containing functional groups were generated on the resin surface [18]. Notably, the XPS spectra of Pd/ABS 0 vol% and Pd/ABS 1 vol% showed a peak at 535.9 eV, which corresponded to the Pd(acac)_2_ peak [22]. This result also supports the formation of Pd coordinated to the oxygen atoms of the carbonyl groups.

### 3.2. Electroless Cu Plating of Pd/ABS

Figure 3 shows the exterior images of the samples after the electroless plating step. All the Pd/ABS samples were completely covered with electroless Cu plating. Therefore, Pd immobilized on ABS substrates served as catalysts for the electroless plating process, regardless of whether the chemical states of Pd were metallic or coordinated to the oxygen of carbonyl groups. The results of the peel strength tests are shown in Figure 4. The peel strength of the Cu film on the Pd/ABS samples was 0.7 N/mm or higher. This result indicates that the Pd/ABS samples prepared in this study had high adhesion strength, despite not undergoing treatments such as etching with chromic acid. Although lower than the peel strength of the Cu film of the control sample prepared via the conventional plating method using hexavalent chromic acid (1.4 N/mm), the peel strengths of the Cu films of the Pd/ABS samples were 0.82, 1.00, and 1.17 N/mm, respectively. The results indicate that, as the amount of 2-propanol increased, the peel strength increased. 

## 4. Discussion

The immobilization processes for Pd nanoparticles induced by electron beam irradiation are now discussed. The radiochemical reactions for the reduction in Pd ions in aqueous solution systems have been previously described, as follows [19]:H_2_O ⟿ e_aq_^−^, H^•^, OH^•^, etc. (1)
R–OH + OH^•^ → R′^•^–OH + H_2_O(2)
R–OH + H^•^ → R′^•^–OH + H_2_(3)
Pd^2+^ + 2e_aq_^−^ → Pd^0^(4)
Pd^2+^ + 2H^•^ → Pd^0^(5)
Pd^2+^ + 2R′^•^–OH → Pd^0^ + 2(R′ = O + H^+^)(6)

Equation (1) represents water radiolysis. The reaction generates H radicals and hydrated electrons, which are strong reductants that are capable of reducing Pd^2+^ ions to form metallic Pd nanoparticles, as shown in Equations (4) and (5). It should be noted that OH radicals are the oxidizing species. In the presence of 2-propanol, the radicals are scavenged to form alcohol radicals, as shown in Equations (2) and (3). Thus, the formed alcohol radicals also reduce the Pd^2+^ ions to form metallic Pd nanoparticles. In contrast, in the absence of 2-propanol, OH radicals are not scavenged and act as oxidizing species. 

As shown in Figure 2a, metallic Pd nanoparticles were obtained from samples synthesized in the presence of 2-propanol. These metallic Pd nanoparticles, which are supposed to be immobilized on ABS substrates to minimize their surface energy, are formed by reactions (4), (5) and (6) in solution [23]. In contrast, in the sample synthesized without 2-propanol, Pd was coordinated to the oxygen of the carbonyl groups, as shown in Figure 2a,b. From the results, it was assumed that carbonyl groups were generated on the ABS surface by the oxidation reactions induced by OH radicals. In the presence of 2-propanol, OH radicals were scavenged by reaction (3), which inhibited the oxidation of ABS. These interpretations explain the XPS results shown in Figure 2b. In summary, electron beam irradiation simultaneously induces the introduction of functional groups on ABS and the reduction in Pd nanoparticles, which leads to the immobilization of Pd nanoparticles. Liu reported the controlled immobilization of nanoparticles by addition of block copolymer brushes on resin substrates [24]. On the contrary, we successfully immobilized Pd nanoparticles directly on ABS substrates without any special chemical pretreatment. Ionizing radiation induces reduction in Pd^2+^ ions and chemical modification of ABS surface simultaneously, which resulted in the immobilization of Pd nanoparticles. However, there are still some problems to be improved. As shown in ICP-AES analysis, the loading amount of Pd on ABS did not depend on the concentration of 2-propanol. Additionally, the distribution of Pd nanoparticles on the ABS surface was not homogeneous. Further study is required to improve the synthesis technique.

Next, the results of the electroless plating are discussed. All the Pd/ABS samples were completely covered after electroless Cu plating (Figure 3). As mentioned previously, Pd/ABS samples were treated with an accelerant solution prior to the electroless plating process. XPS analysis revealed that the Pd on the substrates existed in a metallic state after the acceleration step (Figure 5). These results indicate that Pd species immobilized on the substrates were successfully reduced by the acceleration step to form metallic Pd, which served as catalysts for the electroless plating process. The peel strength of the Cu plating exhibited a practical strength of over 0.7 N/mm. High peel strength was obtained even without etching treatment. This is presumably due to the high adhesive strength between the Pd species and the ABS substrate. The peel strength of the electroless plating increased as the amount of 2-propanol added increased. It was presumed that the load weight and chemical state of Pd contributed to the peel strength. However, Pd/ABS 1 vol% showed a high peel strength even though the load weight of Pd was less than that of Pd/ABS 0 vol% and Pd/ABS 10 vol%. From this result, it was inferred that the chemical states of Pd were largely contributed the adhesive strength of the plating rather than the amount of Pd immobilized on ABS substrates. Further research is required to control the load weight of Pd and to determine degree of contribution the load weight and chemical state of Pd to the adhesive strength of the plating.

## 5. Conclusions

Pd nanoparticles were immobilized on ABS substrates using a radiation-based method. Nanoparticle synthesis and surface modification of the polymer substrates were realized simultaneously. Immobilized Pd nanoparticles were used as electroless plating catalysts. The adhesion strength of the plated film was high despite the absence of etching treatment. This method could be applied for immobilizing metal nanoparticles not only on plastic plates but also on various other materials.

## Figures and Tables

**Figure 1 nanomaterials-12-01463-f001:**
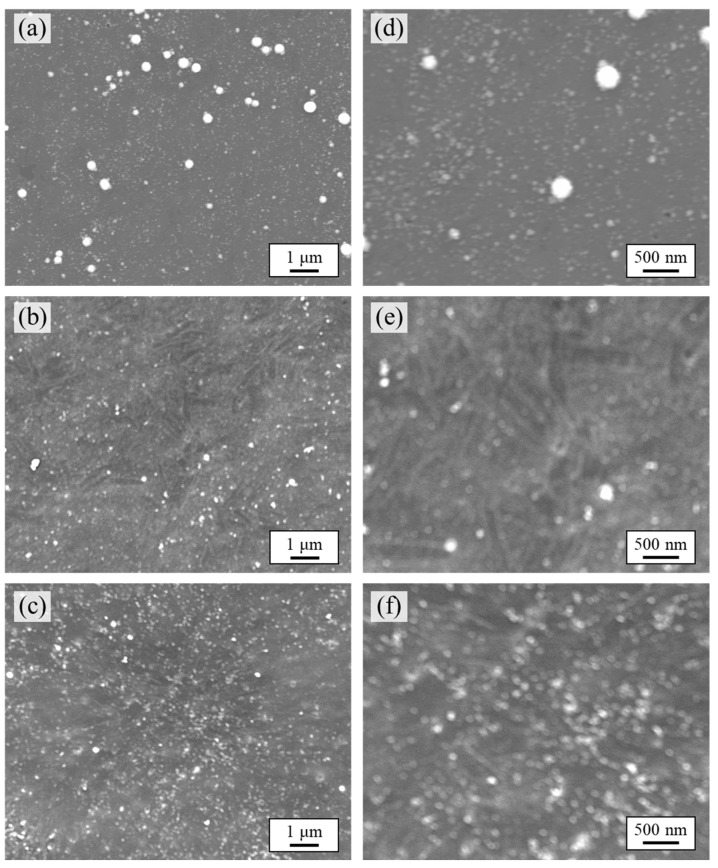
SEM images of Pd/ABS: (**a**–**c**) low-magnification SEM images of Pd/ABS 0 vol%, Pd/ABS 1 vol%, and Pd/ABS 10 vol%; (**d**–**f**) high-magnification SEM images of Pd/ABS 0 vol%, Pd/ABS 1 vol%, and Pd/ABS 10 vol%.

**Figure 2 nanomaterials-12-01463-f002:**
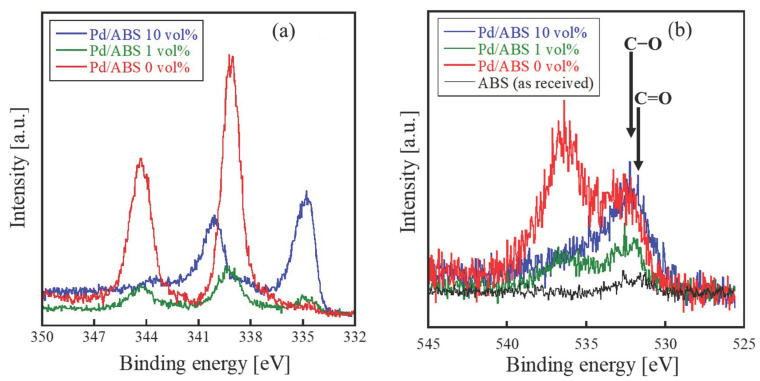
XPS spectra of Pd/ABS: (**a**) Pd3*d*, (**b**) O1*s*.

**Figure 3 nanomaterials-12-01463-f003:**
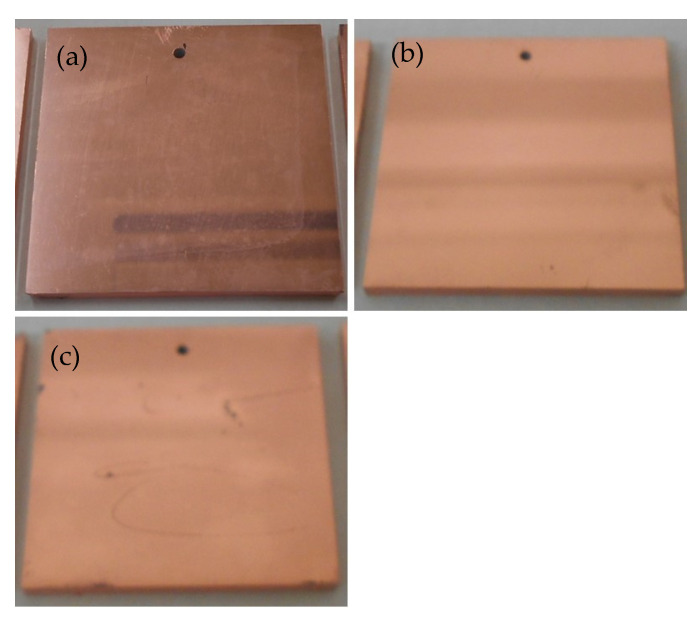
Exterior pictures of Pd/ABS samples after electroless Cu plating: (**a**) Pd/ABS 0 vol%, (**b**) Pd/ABS 1 vol%, and (**c**) Pd/ABS 10 vol%.

**Figure 4 nanomaterials-12-01463-f004:**
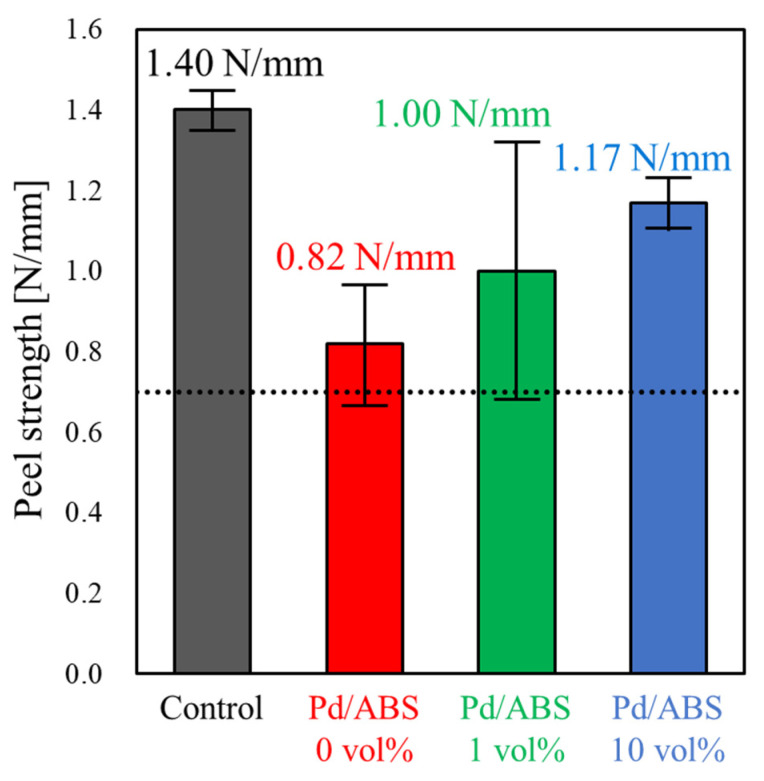
Peel strength of the Cu film on the Pd/ABS. Control sample was prepared by the conventional plating method using hexavalent chromic acid. The dotted line represents 0.7 N/mm.

**Figure 5 nanomaterials-12-01463-f005:**
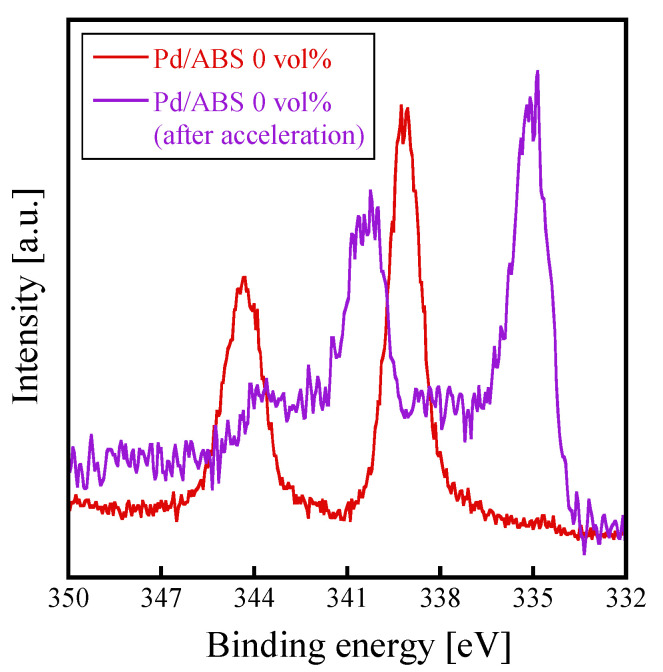
Pd3*d*-XPS spectra of Pd/ABS 0 vol% before and after the acceleration step. The XPS spectrum before the acceleration step (red line) is normalized to a peak intensity of 339.2 eV. The XPS spectrum after the acceleration step (purple line) is normalized to a peak intensity of 335.2 eV.

## Data Availability

The data presented in this study are available on request from the corresponding author.

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
