# Peer review of "Development of a Simultaneous Process of Surface Modification and Pd Nanoparticle Immobilization of a Polymer Substrate Using Radiation"

_nanomaterials, 2022, doi:10.3390/nano12091463_

Round 1
Reviewer 1 Report
Authors reported the synthesis of Pd nanoparticles immobilized on the ABS substrate surfaces by irradiating method, and investigated their morphological, structural and adhesion strength. The results demonstrated that the Pd/ABS samples have high adhesion strength. This method is expected to immobilize metal nanoparticles on plastic plates on various other materials. However, some issues should be addressed.
1, Is the material (Pd/ABS) obtained from physical mixing or in-situ growth? It’s not mentioned in the manuscript. There is no way to judge physical mixing or in-situ growth by the current data. The effect of the interaction to improve the adhesion strength will be neglected if the material just be obtained by physical mixing. In addition, it is better to introduce the existed interaction between Pd and ABS since it is of importance for improving the mechanical performance.
2, In SEM image, the poor resolution and small scale bar of images are difficult for readers to recognize the particle size of approximately 50-200 nm that authors claimed in this work. I suggest authors replace these images by other high-resolution images.
3, I noticed that “These results indicate that 2-propanol used as a radical scavenger, significantly affects the chemical state of the immobilized Pd nanoparticles.” However, when the 1% of 2-propanol was applied, the amounts of Pd nanoparticles immobilized on the resin substrates were 0.3, which was obviously smaller than these of Pd nanoparticles with experimental conditions of 0 and 10 vol% 2-propanol. Please explain these.
4, Controllable and rational processing is a determinant in fabrications of composite. How to improve the controllability and interaction of particle on polymer in this work? The authors should also pay attention to this challenge, and some pioneering and original researches about controllable assembly of nanoparticles are suggested and cited: Giant, 2021, 8, 100076.
- The resolution of Figure 3 is too low and please replace them by high-resolution ones.
6, It is obvious that as the amount of Pd/ABS increased, the peel strengths were also enhanced. I am wondering that since the peel strength does not reaches the peak value, why Pd/ABS samples with the higher amount are not synthesized?
Reviewer 2 Report
The paper addresses the Pd nanoparticles immobilized on an ABS copolymer using ionizing radiation. The results show the Pd/ABS samples have high adhesion strength, and the manuscript has several weaknesses.
- Please provide the sample dimension for the adhesion strength test.
- The authors state that as the amount of 2-propanol increased, the peel strength increased (line 165). The authors also note the high adhesive force between the Pd species and the ABS substrate.
Authors should provide whether the load weight or chemical state of Pd contributes to the adhesive strength of the plating.
- The Pd peaks’ intensity of XPS spectra of Pd/ABS 0 vol% after the acceleration step is weak. Authors should explain why the Pd peaks’ intensity of XPS spectra after the acceleration step is weak?
Round 2
Reviewer 1 Report
All issues were well addressed, and this work can be accepted.
Reviewer 2 Report
The authors answered all questions and comments, and the manuscript was modified systematically.